# Vascular Calcification Is Associated with Fetuin-A and Cortical Bone Porosity in Stone Formers

**DOI:** 10.3390/jpm12071120

**Published:** 2022-07-10

**Authors:** Fernanda Guedes Rodrigues, Rodrigo Fernandes Carvalho Azambuja Neves, Milene Subtil Ormanji, Priscila Ligeiro Gonçalves Esper, Melissa Gaspar, Rosa Maria Rodrigues Pereira, Lucio R. Requião-Moura, Martin H. de Borst, Ita Pfeferman Heilberg

**Affiliations:** 1Nutrition Post Graduation Program, Universidade Federal de São Paulo (UNIFESP), São Paulo 04023-062, Brazil; f.guedes.rodrigues@umcg.nl; 2Department of Nephrology, University Medical Center Groningen, University of Groningen, 9713 GZ Groningen, The Netherlands; m.h.de.borst@umcg.nl; 3Hospital do Rim, Fundação Oswaldo Ramos, São Paulo 04038-002, Brazil; roazambuja@hotmail.com (R.F.C.A.N.); memel_gt@yahoo.com.br (M.G.); lucio.requiao@gmail.com (L.R.R.-M.); 4Nephrology Division, Universidade Federal de São Paulo (UNIFESP), São Paulo 04023-062, Brazil; milene.ormanji@gmail.com (M.S.O.); priscila82@gmail.com (P.L.G.E.); 5Rheumatology Division, Faculdade de Medicina, Universidade de São Paulo (USP), São Paulo 04023-062, Brazil; rosamariarp@yahoo.com

**Keywords:** vascular calcification, nephrolithiasis, kidney stones, bone, fetuin-A, HR-pQCT

## Abstract

Background: Nephrolithiasis has been associated with bone loss and vascular calcification (VC), reflecting abnormal extraosseous calcium deposition. Fetuin-A (Fet-A) acts as a potent inhibitor of ectopic mineralization. The aim of the present study was to evaluate the prevalence of VC in stone formers (SF) and non-stone formers (NSF) and to investigate potential determinants of VC among SF, including circulating levels of Fet-A and bone microarchitecture parameters. Methods: Abdominal aortic calcification (AAC) was assessed using available computed tomography in SF and in age-, sex-, and BMI-matched NSF (potential living kidney donors). Serum Fet-A was measured in stored blood samples from SF. Bone microarchitecture parameters were obtained as a post hoc analysis of a cross-sectional cohort from young SF evaluated by high-resolution peripheral quantitative computed tomography (HR-pQCT). Results: A total of 62 SF (38.0 [28.0–45.3] years old) and 80 NSF (40.0 [37.0–45.8] years old) were included. There was no significant difference in AAC scores between SF and NSF. However, when dividing SF according to mean AAC score, below <5.8% (*n* = 33) or above ≥5.8% (*n* = 29), SF with higher AAC presented significantly higher BMI and tibial cortical porosity (Ct.Po) and significantly lower serum HDL, klotho, Fet-A, and eGFR. Urinary calcium did not differ between groups, but fractional excretion of phosphate was higher in the former. Upon multivariate regression, BMI, serum Fet-A, and tibial Ct.Po remained independently associated with AAC. Conclusions: This study suggests an association between reduced circulating Fet-A levels and increased bone Ct.Po with VC in SF.

## 1. Introduction

Nephrolithiasis represents a common disorder with a lifetime cumulative incidence of 5–10% and a progressively increasing prevalence worldwide [1,2], affecting all ages, sexes, and races, but frequently occurring between the 2nd and 4th decades of life [3]. Lithogenesis is characterized by an imbalance between the solubility and precipitation of minerals in the urine influenced by several factors, including climate, social-economic status, eating habits, obesity, genetic inheritance, and metabolic disorders [2,4,5,6,7,8,9,10,11]. It is well established that stone formers (SF) exhibit a reduction of bone mineral density (BMD) [12,13], which may be exacerbated under low calcium intake [14], and histomorphometric studies have reported low bone formation and increased bone resorption [15,16,17]. Bone demineralization occurs mostly in SF with idiopathic hypercalciuria [16,17,18], but unexpectedly in patients with normal urinary calcium levels as well [19,20,21,22].

In the last decades, numerous studies have shown associations between nephrolithiasis and cardiovascular diseases [23,24,25,26,27,28,29] but the underlying pathophysiological mechanisms remain not fully elucidated. Bone loss and demineralization are common among patients with vascular calcification (VC) [30,31,32], and both calcium nephrolithiasis and VC can be considered as extraosseous sites of abnormal calcium deposition. The Coronary Artery Risk Development in Young Adults (CARDIA) study, a 20-year follow-up retrospective cohort, revealed that young kidney SF had a higher prevalence of subclinical carotid atherosclerosis than NSF [33], disclosing for the first time an association between nephrolithiasis and VC [33]. Subsequently, Shavit et al. [34] reported more severe abdominal aortic calcification (AAC) scores and lower vertebral BMD in SF than age-matched non-SF (NSF) but AAC scores were not associated with hypercalciuria in their series of SF. Other investigators have also observed the presence of VC among SF [27,28,35], but not all [36].

Fetuin-A (Fet-A) is an important regulator of extracellular matrix mineralization, considered as a key systemic mineral chaperone and inhibitor of soft-tissue and VC [37]. In addition, Fet-A plays a critical role in forming and stabilizing high molecular weight colloidal protein–mineral complexes known as calciprotein particles (CPP) [37], which help to transport and to clear mineral nanocrystals from the circulation by the mononuclear phagocytic system [38]. However, hydroxyapatite- and protein-containing CPPs are important drivers of calcification because the amorphous of primary CPP-I transformation into a crystalline or secondary form CPP-II induces vascular smooth cell (VSMC) calcification [39,40]. Some investigators have observed lower serum and urinary Fet-A levels in SF than in controls [41,42]. Nevertheless, the association of Fet-A levels with VC and bone metabolism as another player in the population of SF has not been previously addressed. The aim of the present study was to evaluate the prevalence of VC in SF and NSF and to investigate potential determinants of VC among SF, including circulating levels of Fet-A and bone microarchitecture parameters (tibial and radius trabecular number and separation, and tibial and radius cortical porosity).

## 2. Methods

### 2.1. Study Population

This study is a post hoc analysis of a prospective cohort that evaluated bone microarchitecture parameters by high-resolution peripheral quantitative computed tomography (HR-pQCT) in young SF [21]. The initial sample underwent nonenhanced computed tomography (CT) scans and full metabolic evaluation as a routine care at the nephrolithiasis outpatient clinic of the Nephrology Division of Universidade Federal de São Paulo (UNIFESP). Exclusion criteria was men over 60 years old, postmenopausal women, estimated glomerular filtration rate (eGFR) by the Chronic Kidney Disease Epidemiology Collaboration (CKD-EPI) equation [43] < 90 mL/min/1.73 m^2^, diabetes mellitus, renal tubular acidosis, hyperparathyroidism, or past use of thiazides, corticosteroids, and anticonvulsants. Written consent was obtained from each patient, and the study protocol approved by the local Medical Ethics and Research Committee of UNIFESP (number 4.869.310), in accordance with the Helsinki Declaration of 1975.

In the present prospective study, the presence of AAC was assessed based on available CT imaging exams, and the quantification was calculated according to previous description by other investigators [28,34,44]. Age-, sex-, and BMI-matched healthy NSF (NSF) retrieved from a list of potential living kidney donors served as controls for AAC assessment, as these subjects were submitted to contrast-enhanced CT scans (plus blood biochemistry) as part of their screening for kidney donation at the Kidney Transplant outpatient clinic at Hospital do Rim.

Clinical and anthropometric data, laboratory tests, and smoking habits were retrieved from the medical records of SF and serum total Fet-A was measured in their stored blood samples whenever available.

### 2.2. AAC Score

Each CT slice was scored individually for calcified circumference in between the celiac axis and the aortic bifurcation, and were determined to be either mildly, moderately, or severely calcified (≤10% of circumference as mild, 11–50% as moderate, and ≥51% as severe) [28,34,44]. To provide comparable scoring, a severity multiplier to yield an overall severity score, namely, an AAC score was devised, disposing to each category a severity factor (×5 for mild, ×30 for moderate, and ×75 for severe). These values were representative of the average percentage of circumference calcified in each category. The number of slices in the three severity categories were given as percentages of the total aorta calcified to get rid of errors from different number of slices per patients, and were then multiplied by the respective factor to get an overall AAC severity score.

### 2.3. Serum and Urinary Parameters

Laboratorial parameters included serum creatinine, ionized calcium, phosphate, magnesium, parathyroid hormone (PTH), bone alkaline phosphatase (BAP), 25-hydroxyvitamin D, 1,25-dihydroxyvitamin D3, glucose, sclerostin, C-telopeptide cross-link of type 1 collagen (CTX), Procollagen type 1 N-terminal propeptide (P1NP), klotho, fibroblast growth factor-23 (FGF23), glucose, total cholesterol, HDL and LDL cholesterol, triglycerides, and uric acid. Creatinine levels were determined according to a modified Jaffe’s reaction, using an isotope dilution mass spectrometry (ID-MS) traceable method. Vitamin D metabolites and PTH were determined by chemiluminescent immunoassays (Architect, Abbott, Park, IL, USA), Sclerostin, Bone Alkaline Phosphatase (BAP), FGF23 and klotho by ELISA kits (Teco Medical, Sissach, Switzerland; Cusabio, Houston, TX, USA; Quidel, San Diego, CA, USA; Immutopics Inc., San Diego, CA, USA; IBL, Minneapolis, MN, USA, respectively). P1NP and CTX were measured using the automated electrochemiluminescence immunoassays by Roche Inc. (Tokyo, Japan). Calcium, sodium, urea, and phosphorus were determined in 24-h urine samples. Urinary calcium, urea, and phosphorus were determined by a colorimetric method and sodium by ion-selective electrode. Hypercalciuria was defined as urine calcium excretion ≥ 300 mg/dL for men and ≥ 250 mg/dL for women [45]. All biochemical parameters were measured in a Beckman Clinical Chemistry Analyzer (AU480-America Inc., PA, USA).

Serum total Fet-A was measured using a commercially available ELISA kit (Biovendor, Brno, Czech Republic). Inter-assay imprecision was 5.7% at 30 mg/L and the limit of detection was 0.4 mg/L. For the estimation of Fet-A-containing calciprotein (CPP), aliquots (500 mL) of each serum sample were subjected to further centrifugation for 2 h at 24,000× *g* and 4 °C. The supernatant was then re-analyzed for Fet-A using the same ELISA assay. CPP-containing Fet-A levels were expressed as a percentage of the total serum concentration using the following formula: reduction ratio, Fet-A RR = (serum total Fet-A − supernatant Fet-A)/serum total Fet-A × 100, as described by Smith et al. [46]. The limit of quantification for this analysis was determined to be at least 4.7% [46]. All ELISA measurements were made in duplicate, and the mean concentration was used in subsequent analysis.

### 2.4. High-Resolution Peripheral Quantitative Computed Tomography (HR-pQCT)

Trabecular and cortical microarchitecture were determined in all participants by HR-pQCT (XtremeCT, Scanco Medical AG, Brüttisellen, Switzerland) at the non-dominant distal radius and tibia sites, with a spatial resolution of 82 µm. Variables considered in the present analysis included microarchitecture parameters such as trabecular number (Tb.N), trabecular separation (Tb.Sp), and cortical porosity (Ct.Po).

### 2.5. Statistical Analysis

Statistical analyses were performed using IBM SPSS version 23.0 (SPSS Inc., Chicago, IL, USA). In all analyses, *p* < 0.05 was considered significant. Variables distributions were evaluated by the Kolmogorov–Smirnov test, normally distributed variables as mean ± standard deviation (SD), and non-normally distributed variables as median (interquartile range). Differences between groups were tested by Mann–Whitney U pairwise or Student *t*-test according to their distribution. Categorical variables, presented as *n* (%), were compared using a Chi-square test.

SF were divided into two groups according to the mean value of AAC, in order to identify the main differences between groups. Determinants of AAC Score were further investigated using univariable linear regression, which included potentially relevant clinical and/or laboratorial factors for VC (i.e., age, sex [47], smoking [48], BMI [49], renal function, serum levels of Fet-A [50], sclerostin [51], ionized calcium [50], phosphate [50], PTH, BAP, klotho, FGF23 [52], 25OH and 1-25OH-vitamin D [53], blood lipids [54], urinary calcium, and phosphate [50]). Subsequently, all variables with a *p* < 0.05 were included in a multivariable linear regression model to identify independent determinants of AAC score. Residuals were checked for normality and were natural log-transformed when appropriate.

## 3. Results

From a total of 106 patients, 62 SF (33M/29F, 38.0 [28.0–45.3] years old) whose CT images could be retrospectively accessed, met the inclusion criteria for this study. Table 1 shows their demographic characteristics, laboratory parameters, and AAC score in comparison to 80 age-matched NSF (controls). SF presented significantly higher levels of serum triglycerides but no other statistical differences in laboratory parameters when compared to NSF. A total of 60 out of 62 SF (97.6%) presented mild calcification (<11% of aortic circumference calcified) and 2 (3.4%) presented moderate calcification (11–50% of aortic circumference), with no individual presenting severe scores of calcification (≥51% of aortic circumference calcified). Moreover, all NSF presented mild calcification (data not shown in tables). The score of AAC did not differ between SF and NSF.

Table 2 shows demographic, laboratory, and bone microarchitecture parameters of SF divided according to their mean AAC, above ≥5.8% (*n* = 29) or below <5.8% (*n* = 33). SF with higher AAC also presented significantly higher BMI and a trend for a higher duration of disease (*p* = 0.06), and significantly lower serum HDL, klotho, Fet-A, and eGFR. There were no statistically significant differences in age, presence of hypertension, metabolic syndrome, and smoking. The percentage of hypercalciuria did not differ between both groups, nor did the urinary calcium levels, but fractional excretion of phosphate was higher in the former group. Interestingly, regarding the HR-pQCT parameters, SF with higher AAC score also showed higher tibial cortical porosity (*p* < 0.05), although the other bone parameters were not significantly different between groups. Other 24 h lithogenic urinary parameters are shown in Appendix A. AAC score was positively correlated with 24 h urinary phosphate and fractional excretion of phosphate, however it did not correlate with calcium excretion (Appendix A). CPP-containing Fet-A levels, expressed as a percentage of the total serum concentration (serum Fet-A RR), detectable by values higher than >4.7% [46], were observed in 66.6% of SF (data not shown in tables).

Table 3 shows potential determinants of AAC score as a dependent variable. Univariate linear regression revealed associations with smoking (β 0.29, *p* = 0.02), BMI (β 0.32, *p* = 0.01), serum sclerostin (β 0.30, *p* = 0.02), tibial cortical porosity (β 0.30, *p* = 0.02), and urinary phosphate (β 0.36, *p* < 0.01), and inverse association with eGFR (β −0.31, *p* = 0.01), serum 25OH-vitamin D (β −0.24, *p* = 0.05), and serum Fet-A (β −0.35, *p* < 0.01). However, upon multivariate analysis, only BMI, serum Fet-A, and tibial cortical porosity remained independently associated.

## 4. Discussion

To the best of our knowledge, the present study is the first to investigate the association between VC, bone microarchitecture evaluated by HR-pQCT, and serum Fet-A levels in a relatively young SF population. Unlike previous studies, in addition to controlling for age and gender with healthy controls, BMI was included in the matching process and medical comorbidities linked to calcification such as diabetes mellitus were excluded from the sample of SF. Herein, around 98% of SF exhibited only mild abdominal aortic calcification AAC scores with no statistical differences compared to age, sex, and BMI-matched NSF. However, within the group of SF, AAC score was positively associated with BMI and higher tibial cortical porosity, while inversely associated with serum Fet-A levels.

Notably, CPP expressed as serum Fet-A RR was detectable in 66.6% of SF, although it did not differ between SF with AAC values higher or lower than 5.8%. Given that under physiological conditions CPPs should be rapidly cleared and not detectable in the circulation [46,55], their presence may reveal that either the rate of formation was increased or removal rate was reduced or at least exceeded the capacity of the clearance pathway [46]. A handful of preceding studies searched for possible differences between SF and controls regarding AAC [28,34,35,36] or aortic coronary calcification [27] with distinct results. Present findings showing no statistical differences between SF and non-SF with respect to AAC scores prevalence are in line with previous data from Shavit et al. [34], although a greater degree of severity of AAC scores among SF was disclosed by these investigators.

When clustering the SF into two groups regarding the AAC score, in the subgroup of patients with higher VC (AAC > 5.8%), there was a male preponderance (although without reaching statistical significance, *p* = 0.07), as observed in a previous study assessing aortic calcification scores where higher values were seen in 20–40 years old SF males compared to controls [28]. The group with AAC > 5.8% also presented significantly higher BMI and serum uric acid, with lower HDL and eGFR, but the percentage of patients with metabolic syndrome was not different from the group with AAC < 5.8%. Patients who presented higher AAC score had a trend for longer history of stone disease (*p* = 0.06).

Interestingly, the former group exhibited a significantly higher fractional excretion of phosphate (with no differences in serum phosphate) accompanied by lower levels of serum klotho, and a trend for higher serum FGF-23 (*p* = 0.06), albeit still within the normal range. Klotho deficiency is associated with reduced renal function, hyperphosphatemia, increased FGF23 levels, renin–angiotensin–aldosterone system (RAAS) activation, inflammation, and chronic complications such as ectopic calcification, among others [56,57].

Despite of some controversy regarding the potential effects of FGF23 and klotho on VC [52], their contribution to ectopic calcification in soft tissues in addition to traditional factors is thus given due consideration [58], especially under conditions of CKD [59]. Altogether, as renal function is slightly decreased but still well preserved in the SF subgroup with AAC > 5.8%, it is possible to speculate about the increased fractional excretion of phosphate as an attempted secondary defense mechanism against further VC. Renal phosphate leak has been described among calcium nephrolithiasis patients [60] but in the current series, only one patient presented mild hypophosphatemia (data not shown). Calciprotein monomers (and not primary CPP containing amorphous calcium phosphate or secondary crystalline CPPs) are potent inducers of FGF23 production implying that Fet-A is required for FGF23 secretion [55,61]. However, as suggested by Jahnen-Dechent et al. [37], Fet-A-deficient mice greatly increased, not suppressed FGF23, as also observed in the present study.

Ferraro et. al. [35] observed that urinary stones whose composition revealed calcium phosphate content in higher proportion were directly associated with AAC, suggesting a common pathway involving ectopic calcification in the kidney and vessels as an abnormal biomineralization process and mineral crystallization [62]. However, as no urinary stone composition analysis data were available in the present study, we could not address whether AAC scores were related to a different calcium salt composition.

Although serum sclerostin levels were not different between groups, they were positively associated with AAC score in univariate linear regression but did not remain significant upon multivariate adjustment. Since circulating sclerostin has been associated with VC mainly in a CKD setting [63], the present result might be explained by normal kidney function in the currently studied population. Nevertheless, this relationship still remains controversial with negative or null associations in distinct populations [64,65,66].

So far, no study has accessed associations between bone microarchitecture assessed by HR-pQCT and VC in SF. Nephrolithiasis was systematically associated with low bone mineral density [12,13] and increased fracture risk [67], and previous histomorphometric analysis by our group showed high bone resorption and a mineralization defect [16,17]. More recently, we detected trabecular bone impairment and reduced bone strength by HR-pQCT in SF when compared to healthy controls [21]. Here, the subgroup of patients with higher VC (AAC > 5.8%) also exhibited significantly higher tibial cortical porosity, which may be consequent to bone resorption. As lately suggested by Jahnen-Dechent and Smith [55] minerals (i.e., calcium and phosphate) and Fet-A can be released from the bone matrix by the action of osteoclasts during bone resorption, consequently increasing the formation of calciprotein particles [55]. The current lower serum Fet-A levels could be due to its consumption by the process of CPP formation secondary to the higher bone cortical porosity found in the group with AAC > 5.8%. In the present series, the finding of an independent association between tibial cortical porosity (Ct.Po) and AAC in the multivariate linear regression analysis suggests this underlying mechanism, but, since serum Fet-A RR, as a proxy for CPP, was not different between AAC score groups, such hypothesis could not be confirmed. Of note, an experimental study showed that cortical rather than trabecular bone loss was correlated with the severity of VC in CKD rats [68]. Although Shavit et al. [34] and Fabris et al. [29] did not perform association analysis of VC and bone demineralization, they proposed the existence of common underlying pathways leading to increased extraosseous calcium depositions in kidneys and blood vessels.

Furthermore, the multivariate linear analysis did reveal an inverse association between AAC score and serum Fet-A, in accordance with the literature [69], which defines Fet-A as a serum component that prevents vascular smooth muscular cells (VSMCs) by binding hydroxyapatite vesicles and therefore reducing its calcification potential [69]. As aforementioned, several studies have shown reduction of serum Fet-A levels in SF [41,42,70]. As Fet-A deficiency is also associated with increased cardiovascular mortality and coronary and valvular calcification in patients with ESKD [71,72], it is conceivable that reduced concentrations of such calcification inhibitor may explain, at least in part, the apparent link between nephrolithiasis and VC. The direct association of BMI associated with AAC score among SF in the multivariate analysis, is in accordance with findings from other investigators [49].

Nevertheless, our study has several limitations. First, due to the cross-sectional nature of our study sample, it is not possible to prove causalities. This is a single-center study, and thus the results could not be generalized to the overall population. The relatively small sample size in this study has to be acknowledged, limiting extensive multivariate correction for possible confounders, and there has been no urinary stone composition analysis available. The absence of stored blood samples from the healthy kidney donors used as NSF controls did not allow us to determine circulating Fet-A and specific serum parameters other than routine biochemistry. In order to overcome the limitation of applying a gold standard methodology to quantify calciprotein particles (CPP), as described by Pasch et al. [73], we performed the indirect analysis of the apparent reduction in serum Fet-A concentration (reduction ratio, RR) after high-speed centrifugation [46]. However, as opposed to Pasch et al. [73], who investigated hemodialysis patients versus healthy subjects (HS), with expected differences regarding serum calcium and phosphate levels, stone formers (SF) with normal renal function did not present altered levels of such parameters in the present study, rendering the comparison of our results not applicable.

Notwithstanding, statistically significant findings presented here despite the small sample size underscore the biological relevance of the findings. Strengths of this research include the novelty of evaluation of circulating Fet-A, HR-pQCT, and VC in SF.

## 5. Conclusions

In conclusion, our study suggests that despite a small prevalence of abdominal aortic calcification scores among relatively young SF, an association between circulating Fet-A levels and bone cortical porosity with VC was disclosed. Such findings may represent an additional potential link between nephrolithiasis, bone, and cardiovascular disease, warranting further clinical trials involving a larger number of patients and other populations.

## Figures and Tables

**Table 1 jpm-12-01120-t001:** Demographic characteristics, laboratory parameters, and AAC score (%) of SF and NSF.

	SF (*n* = 62)	NSF (*n* = 80)	*p* Value
Sex (Male), *n* (%)	29 (42.7)	29 (36.3)	0.14
Caucasian, *n* (%)	34 (54.8)	49 (61.3)	0.22
Afro-Brazilian, *n* (%)	27 (43.5)	31 (38.8)	
Asian, *n* (%)	1 (1.6)	0 (0)	
Age, years	38.0 (28.0–45.3)	40.0 (37.0–45.8)	0.10
BMI, kg/m^2^	26.6 ± 4.5	26.8 ± 3.7	0.81
Normal weight, *n* (%)	21 (33.9)	19 (23.8)	0.50
Overweight, *n* (%)	27 (43.5)	35 (43.8)	
Obese, *n* (%)	14 (22.6)	17 (21.3)	
Total cholesterol, mg/dL	190.3 ± 34.3	187.9 ± 36.3	0.37
LDL cholesterol, mg/dL	111.2 ± 27.0	110.4 ± 76.2	0.47
HDL cholesterol, mg/dL	51.6 ± 13.5	52.3 ± 15.9	0.40
Triglycerides, mg/dL	112.0 (77.0–163.0)	84.0 (66.3–127.3)	<0.05
Glucose, mg/dL	90.9 ± 10.3	93.0 ± 11.0	0.16
eGFR, mL/min/1.73 m^2^	125.1 ± 27.1	118.8 ± 20.6	0.10
AAC Score, %	5.8 ± 0.8	5.6 ± 0.7	0.27

Abbreviations: SF, stone formers; NSF, non-stone formers; AAC, abdominal aortic calcification; BMI, body mass index; LDL, low density lipoprotein; HDL, high density lipoprotein; eGFR, estimated glomerular function rate.

**Table 2 jpm-12-01120-t002:** Demographic, laboratorial, and bone microarchitecture parameters of SF divided according to their mean AAC.

	Total *n* = 62	G1 AAC < 5.8% *n* = 33	G2 AAC ≥ 5.8% *n* = 29	*p* Value
AAC Score, %	5.8 ± 0.8	5.2 ± 0.4	6.5 ± 0.6	<0.001
Age, years	38.0 (28.0–45.3)	38.0 (27.5–44.5)	38.0 (31.0–47.5)	0.49
Male, *n* (%)	33 (53.2)	12 (36.4)	17 (58.6)	0.07
BMI, kg/m^2^	26.3 ± 4.5	25.5 ± 4.0	28.0 ± 4.6	0.03
eGFR, mL/min/1.73 m^2^	99.3 ± 13.4	103.2 ± 12.9	94.9 ± 12.7	0.01
Hypertension, *n* (%)	9 (14.5)	6 (18.2)	3 (10.3)	0.30
Metabolic syndrome, *n* (%)	16 (25.8)	7 (21.2)	9 (31.0)	0.32
Smoking, *n* (%)	6 (9.7)	4 (12.1)	2 (6.9)	0.40
Duration of disease, years	7.0 (2.0–14.0)	5.5 (1.0–12.5)	10.5 (5.0–17.3)	0.06
**Serum parameters**				
Creatinine, mg/dL	0.89 (0.75–1.09)	0.77 (0.64–0.90)	0.90 (0.79–1.06)	<0.01
Ionized calcium, mmol/L	1.30 ± 0.04	1.30 ± 0.04	1.31 ± 0.03	0.35
Phosphate, mg/dL	3.3 ± 0.4	3.3 ± 0.4	3.2 ± 0.5	0.35
Magnesium, mg/dL	2.0 ± 0.2	2.1 ± 0.2	2.1 ± 0.2	0.86
Glucose, mg/dL	90.9 ± 10.3	91.8 ± 10.2	89.8 ± 10.5	0.46
Total cholesterol, mg/dL	190.3 ± 34.3	188.6 ± 36.8	192.2 ± 31.8	0.70
LDL cholesterol, mg/dL	111.2 ± 27.0	107.2 ± 19.2	115.7 ± 24.1	0.25
HDL cholesterol, mg/dL	50.0 (41.0–58.0)	54.0 (50.0–61.3)	49.0 (36.0–55.0)	0.03
Triglycerides, mg/dL	95.0 (71.0–151.0)	110.5 (67.8–161.3)	126.0 (80.0–177.0)	0.10
Uric Acid, mg/dL	5.1 ± 1.2	4.4 ± 1.1	5.8 ± 1.5	<0.01
PTH, pg/mL	51.0 (40.0–58.0)	49.0 (41.5–64.5)	53.0 (38.0–73.0)	0.61
25(OH)-vitamin D, ng/mL	25.0 (21.0–30.0)	26.0 (21.0–31.5)	22.5 (19.3–30.3)	0.21
1-25(OH)-vitamin D, pg/mL	23.8 (17.8–35.9)	22.2 (15.0–33.3)	23.8 (17.3–35.9)	0.33
Sclerostin, pmol/L	24.1 (18.9–30.5)	21.7 (18.3––26.7)	23.5 (18.4–30.6)	0.28
CTX, ng/mL	0.38 ± 0.17	0.41 ± 0.19	0.48 ± 0.18	0.94
P1NP, ng/mL	57.8 ± 20.2	62.1 ± 22.8	68.0 ± 22.8	0.46
BAP, U/L	13.8 ± 3.7	13.1 ± 3.6	14.8 ± 3.7	0.08
Klotho, pg/mL	725 (542–941)	854 (584–1355)	575 (393–893)	0.01
FGF-23, pg/mL	33.6 (25.6–39.8)	29.2 (20.4–38.5)	36.0 (25.4–45.0)	0.06
Total Fetuin-A, ug/mL	666.1 ± 102.4	695.8 ± 88.4	633.2 ±108.2	0.02
Serum Fet-A RR (%)	8.6 (4.4–18.1)	8.5 (4.–19.9)	8.4 (4.5–17.2)	0.82
**Urinary parameters**				
Calcium, mg/24 h	230.5 (147–301)	256 (143–300)	221 (140–289)	0.92
Phosphate, mg/24 h	859.1 ± 273.8	796.0 ± 245.9	928.9 ± 290.2	0.06
FeP, %	14.0 (11.3–17.9)	11.8 (9.9–16.7)	13.9 (11.5–18.2)	0.04
**HR-pQCT parameters**				
Tibial Tb.N, 1/mm	1.74 (1.59–2.09)	1.68 (1.55–1.94)	1.70 (1.56–2.22)	0.23
Tibial Tb.Sp, mm	0.50 (0.40–0.55)	0.52 (0.44–0.57)	0.50 (0.37–0.56)	0.17
Tibial Ct.Po, %	0.032 (0.022–0.042)	0.028 (0.021–0.036)	0.035 (0.028–0.041)	<0.05
Radius Tb.N, 1/mm	2.03 (1.82–2.21)	2.03 (1.82–2.21)	1.98 (1.81–2.21)	0.67
Radius Tb.Sp, mm	0.42 (0.34–0.47)	0.43 (0.39–0.48)	0.43 (0.36–0.48)	0.35
Radius Ct.Po, %	0.014 (0.009–0.022)	0.011 (0.007–0.019)	0.015 (0.010–0.022)	0.20

Abbreviations: SF, stone formers; AAC, abdominal aortic calcification; BMI, body mass index; eGFR, estimated glomerular function rate; LDL, low density lipoprotein; HDL, high density lipoprotein; PTH, parathyroid hormone; CTX, C-terminal telopeptide; P1NP, procollagen type I N-terminal propeptide; BAP, bone alkaline phosphatase; FGF-23, fibroblast growth factor 23; FeP, fractional excretion of phosphate; HR-pQCT, high-resolution peripheral quantitative computed tomography; Tb.N, trabecular number; Tb.Sp, trabecular separation; Ct.Po, cortical porosity.

**Table 3 jpm-12-01120-t003:** Linear regression using AAC score as dependent variable.

Potential Determinants	Univariable		Multivariable *	
	B	*p*	B	*p*
Age, years	0.17	0.16	-	-
Sex, F	−0.21	0.10	-	-
BMI, kg/m^2^	0.32	0.01	**0.31**	**<0.01**
Metabolic syndrome, yes	0.22	0.09	-	-
Hypertension, yes	0.16	0.22	-	-
Smoking, yes	0.29	0.02	0.13	0.30
Serum sclerostin, pmol/L	0.30	0.02	0.09	0.50
Tibial Tb.N, 1/mm	0.14	0.27	-	-
Tibial Tb.Sp, mm	−0.15	0.26	-	-
Tibial Ct.Po, %	0.30	0.02	**0.26**	**0.03**
Tibial Ct.Th, mm	0.18	0.17	-	-
Radius Tb.N, 1/mm	0.07	0.59	-	-
Radius Tb.Sp, mm	−0.11	0.40	-	-
Radius Ct.Po, %	0.21	0.10	-	-
Radius Ct.Th, mm	−0.02	0.86	-	-
Urinary calcium, mg/24 h	0.09	0.49	-	-
Urinary phosphate, mg/24 h	0.36	<0.01	0.08	0.60
FeP, %	0.27	0.04	−0.03	0.87
eGFR, mL/min/1.73 m^2^	−0.31	0.01	−0.25	0.24
Serum ionized calcium, mmol/L	0.10	0.40	-	-
Serum phosphate, mg/24 h	0.14	0.27	-	-
Serum 25OH-vitamin D, ng/mL	−0.24	<0.05	–0.03	0.80
Serum 1-25OH-vitamin D, pg/mL	0.07	0.55	-	-
Serum PTH, pg/mL	−0.04	0.78	-	-
Serum BAP, U/L	0.18	0.15	-	-
Serum klotho, pg/mL	−0.26	0.07	-	-
Serum FGF23, pg/mL	0.16	0.20	-	-
Serum Fetuin-A, ug/mL	−0.35	<0.01	**−0.29**	**0.02**
Serum Fetuin-A RR, %	0.15	0.26	-	-

* Run backwards. Abbreviations: AAC, abdominal aortic calcification; BMI, body mass index; Tb.N, trabecular number; Tb.Sp, trabecular separation; Ct.Po, cortical porosity; FeP, fractional excretion of phosphate; eGFR, estimated glomerular function rate; PTH, parathyroid hormone; BAP, bone alkaline phosphatase; FGF-23, fibroblast growth factor 23.

## Data Availability

The data in this article are available from the corresponding author upon reasonable request.

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
