# Peer review of "Vascular Calcification Is Associated with Fetuin-A and Cortical Bone Porosity in Stone Formers"

_jpm, 2022, doi:10.3390/jpm12071120_

Round 1

Reviewer 1 Report

Comments and Suggestions for Authors

The manuscript is an original article that addresses the assessment of serum Fetuin-A, different serum parameters related to bone turnover, bone microarchitecture parameters, and abdominal aortic calcification (AAC) in stone formers (SF) versus non-stone formers (NSF).  The manuscript is well written in terms of the English language. The topic is interesting for researchers and clinicians, providing sufficient background to understand its message. However, it requires improvement by reviewing a few major and minor issues:

Major

1.       Please explain why you used severity multipliers for the AAC score. This artificially increases the differences between AAC scores and could result in statistically significant differences between certain parameters.

The discussions are speculative, and require reformulation or new approaches, as follows:

2.       Regarding calciprotein particles (CPP), lines 232-238: Aren't the CPP changes due to their approximation by formula, and not by their direct assessment by the nephelometric method? (see Pasch et al. 2012, doi: 10.1681/ASN.2012030240). This could be another weakness of the study.

3.       Sclerostin, as a major inhibitor of bone formation, has not been addressed at all, especially since AAC has been associated with sclerostin in univariate linear regression. Why would you quantify it if you didn't approach it anywhere?

4.       Since dietary phosphate intake has not been evaluated, discussions about phosphaturia/-emia are purely speculative. Please remove them, or at least reduce them. This would be another weakness of the study.

5.       Also, discussions about the association of klotho and inflammation are purely speculative since no inflammatory markers were assessed in the study (lines 284-290). Please approach klotho changes only from the point of view of the examinations performed in the study.

6.       Lines 297-298: “…..vascular calcification in the nephrolithiasis setting is a matter of time”. You cannot state this unless you have follow-up study references.

7.       Lines 310-312: you cannot issue this hypothesis since no significant difference was found between fetal-A RR (as a proxy for CPP) in those with AAC <5.8% and> 5.8%, respectively (see Table 2).

8.       Since no urinary stone composition analysis was performed, how do we know that the differences within the SF subgroups according to AAC are not caused by the different compositions of the stones? Especially since AAC did not differ significantly between the SF and NSF groups. Please explain and insert the explanation in the discussions.

Minor

1.       Try to be consistent with the abbreviations throughout the text (e.g. vascular calcification, stone formers, non-stone formers, fetuin-A). Once you have abbreviated them, use only the abbreviations in the manuscript.

2.       There are inconsistencies in the format of the references (line 106), or you forgot to put them in square brackets (lines 307, 323).

1.       Line 55: Not all stones have calcium in their composition. So, please specify that only for those who have stones with Ca in the composition, the statement is valid.

2.       Line 77: Please specify which bone microarchitecture parameters you assessed.

Author Response

REPORT 1

The manuscript is an original article that addresses the assessment of serum Fetuin-A, different serum parameters  related  to  bone  turnover,  bone  microarchitecture parameters, and  abdominal aortic calcification (AAC) in stone formers (SF) versus non-stone  formers  (NSF). The  manuscript  is  well  written  in  terms  of  the  English language. The topic is interesting for researchers and clinicians, providing sufficient background  to  understand  its  message. However,  it  requires  improvement  by reviewing a few major and minor issues:

Major

1.Please  explain  why  you  used  severity  multipliers  for  the  AAC  score.  This artificially increases  the  differences  between  AAC  scores  and  could  result  in statistically significant differences between certain parameters.

We thank the reviewer for the question. As mentioned in the text, we have assigned each category a severity factor (x5 for mild, x30 for moderate, and x75 for severe), based on previous literature description by both Shavit et al 1 and Leckstroem et al 2 This methodology has been chosen in order to eliminate errors from different number of computed tomography (CT) slices exhibiting different percentage of calcified circumference per patient, as described by Leckstroem et al 2. The number of CT slices varied mildly for each patient according to their size (height, weight). The table below illustrates our results.

SF (n=62)

NSF (n=80)

Median (IQR) number of calcified CT slices

113 (105 - 124)

114 (106 - 122)

Number of pts. with circumferences with mild calcification (< 10%) 

60 (96.7%)

80 (100%)

Number of pts. with circumferences with moderate calcification (11-50%)

2 (3.3%)

0

Number of pts. with circumferences with severe calcification (>50%)

0

0

For example, there were 2 patients with 100 CT slices and 20 of them presenting calcified circumferences. Patient A presented with 10 circumferences <10% calcified and 10 with 25% calcified. Patient B presented with 20 circumferences calcified more than 25%. As shown below, it would be not possible to distinguish the severity degree of the calcification if not multiplied  by the abovementioned severity factors:

Patient A: (10*5) + (10*30) = 350/100 = 3.5%

Patient B: (20*30 = 600/100 = 6%

As the same multiplier has been applied to each group and most of the population study presented with only mild calcification, we expected  a minimal impact on statistical results.

The discussions are speculative, and require reformulation or new approaches, as follows:

2.Regarding calciprotein particles (CPP), lines 232-238: Aren't the CPP changes due to their approximation by formula, and not by their direct assessment by the nephelometric method? (see Pasch et al. 2012, doi: 10.1681/ASN.2012030240). This could be another weakness of the study.

The reviewer is right. We recognize the limitations regarding the current methodology, which has been chosen due to technical difficulties on applying the nephelometric method as described by Pasch et al., 2012 3. In order to overcome the application of such gold standard methodology to quantify calciprotein particles (CPP), we have performed the indirect analysis of the apparent reduction in serum Fet-A concentration (reduction ratio, RR) after high-speed centrifugation, as described by Smith et al 4 . Of note, as opposed to Pasch et al.3, who have investigated hemodialysis patients versus healthy subjects (HS), with expected differences regarding serum calcium and phosphate levels, stone formers (SF) with normal renal function did not present altered levels of such parameters in the present study.  Anyway, the disclosure of 66.6% of the SF presenting with detectable Fet-A RR values seemed to represent an important finding, although the latter were not compared between stone formers and HS. In order to emphasize this weakness, in the revised manuscript, we have added it to the limitations paragraph of the discussion session and cited this important reference.

3.Sclerostin, as a major inhibitor of bone formation, has not been addressed at all, especially since AAC has been associated with sclerostin in univariate linear regression. Why would you quantify it if you didn't approach it anywhere?

We agree with the reviewer that the results regarding sclerostin deserve mention in the discussion. We found a positive association in the univariate analysis, which was opposite to our expectation, while the association did not persist upon multivariable adjustment. We hypothesize that the present result might be explained by normal kidney function in the currently studied population, since circulating sclerostin has been associated with VC mainly in the CKD setting.

4.Since  dietary  phosphate  intake  has  not  been  evaluated,  discussions  about phosphaturia/-emia are purely speculative. Please remove them, or at least reduce them. This would be another weakness of the study.

The reviewer is right and we have modified this part of the text accordingly.

5.Also, discussions about the association of klotho and inflammation are purely speculative since no inflammatory markers were assessed in the study (lines 284-290).  Please  approach  klotho  changes  only  from  the  point  of  view  of  the examinations performed in the study.

The reviewer is right and we have shortened this part of the text accordingly.

6.Lines  297-298: “.....vascular calcification in the nephrolithiasis setting is a matter of time”. You cannot state this unless you have follow-up study references.

We agree with the reviewer. The sentence was moved to the third paragraph of the discussion session and was rephrased withdrawing the last part (“…setting is a matter of time”).

7.Lines 310-312: you cannot issue this hypothesis since no significant difference was found between fetal-A RR (as a proxy for CPP) in those with AAC <5.8% and> 5.8%, respectively (see Table 2).

The reviewer is right. We have rephrased the sentence in order to refute the hypothesis.

8.Since no urinary stone composition analysis was performed, how do we know that the differences within the SF subgroups according to AAC are not caused by the  different  compositions  of  the  stones?  Especially  since  AAC  did  not  differ significantly  between  the  SF  and  NSF  groups.  Please  explain  and  insert  the explanation in the discussions.

As suggested, we have inserted a statement that we cannot address whether SF subgroups according to AAC were caused by differences in stone composition in the 7th paragraph of the discussion.

Minor

1.Try to be consistent with the abbreviations throughout the text (e.g. vascular calcification,  stone  formers,  non-stone  formers,  fetuin-A).  Once  you  have abbreviated them, use only the abbreviations in the manuscript.

Thank you for the suggestion. We have revised the manuscript and the abbreviations are consistent now.

2.There  are  inconsistencies  in  the  format  of  the  references  (line  106),  or  you forgot to put them in square brackets (lines 307, 323).

Thank you for the suggestion. We have revised the manuscript and the references format is consistent now.

  1. Line 55: Not all stones have calcium in their composition. So, please specify that only for those who have stones with Ca in the composition, the statement is valid.

We thank the reviewer for the suggestion, we have specified in the text that calcium nephrolithiasis and vascular calcification can be considered extraosseous sites of calcium deposition.

  1. Line 77: Please specify which bone microarchitecture parameters you assessed.

We thank the reviewer for the suggestion, the studied bone microarchitecture parameters were added in the text.

Obs: Please note that due to reordering the text and adding new references, the reference numbers might have changed.

Reviewer 2 Report

Rodrigues et al aimed to investigate the potential determinate as inhibitors of vascular calcification, as Fetuin A, in stone formers. 

Major Comments:

1) Introduction: As you mentioned several studies already investigate the presence of vascular calcification in stone formers (ref. 34, 27, 28, 35, 36). It is not clear, which gap the current study might to close. 

2) What is the Fetuin A concentration in NSF?

3) Detection of differences in CPP radius and serum calcium propensity score (t50) would support your hypotheses of Fetuin A relevance. 

Minor Comments

- Abbrevariations: Please check the consistent use from the first introduction (e.g. Vascular calcification - VC)

- Reference list: Please check ref 52 and 58 - equal !

- line 277-280: Sentence not clear.

- Table 1: Please give all race.

- line 323: Is the number 72 the references (brackets are missing)?

Author Response

Rodrigues et al aimed to investigate the potential determinate as inhibitors of vascular calcification, as Fetuin A, in stone formers. 

Major Comments:

1) Introduction: As you mentioned several studies already investigate the presence of vascular calcification in stone formers (ref. 34, 27, 28, 35, 36). It is not clear, which gap the current study might to close. 

We thank the reviewer for the question. Despite of several studies investigating either vascular calcification or fetuin-A in stone formers, none of them have studied a potential association between  both. Moreover, the possible interplay with bone disease as a third part has yet not been previously addressed. A new sentence at the end of the introduction session has been added to the present revised form of the manuscript, in order to make it more clear.

2) What is the Fetuin A concentration in NSF?

We did not have stored blood samples from NSF group to measure the fetuin-A and other markers. We agree with the reviewer that this could represent a limitation of our study. Therefore, in the revised form of the manuscript we have added it to the limitations paragraph.

3) Detection of differences in CPP radius and serum calcium propensity score (t50) would support your hypotheses of Fetuin A relevance. 

We recognize the limitations regarding the current methodology, which has been chosen due to technical difficulties on applying the nephelometric method as described by Pasch et al.3. In order to overcome the application of such gold standard methodology to quantify calciprotein particles (CPP), we have performed the indirect analysis of the apparent reduction in serum Fet-A concentration (reduction ratio, RR) after high-speed centrifugation, as described by Smith et al. 4 . Anyway, the disclosure of 66.6% of the SF presenting with detectable Fet-A RR values seemed to represent an important finding. In order to emphasize this weakness, in the revised manuscript, we have added such comments in the limitations paragraph of the discussion session and mentioned this important bibliographic reference.  

Minor Comments

- Abbrevariations: Please check the consistent use from the first introduction (e.g. Vascular calcification - VC)

Thank you for the suggestion. We have revised the manuscript and the abbreviations are consistent now.

- Reference list: Please check ref 52 and 58 - equal !

We have corrected the references, thank you.

- line 277-280: Sentence not clear.

We agree with the reviewer, the sentence was removed from the text.

- Table 1: Please give all race.

We have included afro-Brazilians and Asians in the table.

- line 323: Is the number 72 the references (brackets are missing)?

Yes, we have corrected the references.

 Obs: Please note that due to reordering the text and adding new references, the reference numbers might have changed.

REFERENCES

  1. Shavit L, Girfoglio D, Vijay V, et al. Vascular calcification and bone mineral density in recurrent kidney stone formers. Clin J Am Soc Nephrol. 2015;10(2):278-285. doi:10.2215/CJN.06030614
  2. Leckstroem DC, Bhuvanakrishna T, McGrath A, Goldsmith DJ. Prevalence and predictors of abdominal aortic calcification in healthy living kidney donors. Int Urol Nephrol. 2014;46(1):63-70. doi:10.1007/s11255-013-0485-0
  3. Pasch A, Farese S, Gräber S, et al. Nanoparticle-based test measures overall propensity for calcification in serum. Journal of the American Society of Nephrology. 2012;23(10):1744-1752. doi:10.1681/ASN.2012030240
  4. Smith ER, Cai MM, McMahon LP, et al. Serum fetuin-A concentration and fetuin-A-containing calciprotein particles in patients with chronic inflammatory disease and renal failure. Nephrology. 2013;18(3):215-221. doi:10.1111/nep.12021

Round 2

Reviewer 1 Report

The authors responded to all the comments and suggestions I made. I appreciate the effort and promptness of the authors to make the suggested changes. For my part, I believe that the article can be published.

Reviewer 2 Report

Thanks for addressing the concerns.